# An In Silico Methodology That Facilitates Decision Making in the Engineering of Nanoscale Protein Materials

**DOI:** 10.3390/ijms23094958

**Published:** 2022-04-29

**Authors:** Eloi Parladé, Eric Voltà-Durán, Olivia Cano-Garrido, Julieta M. Sánchez, Ugutz Unzueta, Hèctor López-Laguna, Naroa Serna, Montserrat Cano, Manuel Rodríguez-Mariscal, Esther Vazquez, Antonio Villaverde

**Affiliations:** 1CIBER de Bioingeniería, Biomateriales y Nanomedicina (CIBER-BBN), C/Monforte de Lemos 3-5, 28029 Madrid, Spain; eric.volta@uab.cat (E.V.-D.); jsanchezqa@gmail.com (J.M.S.); uunzueta@santpau.cat (U.U.); hector.lopez@uab.cat (H.L.-L.); esther.vazquez@uab.es (E.V.); 2Institut de Biotecnologia i de Biomedicina, Universitat Autònoma de Barcelona, 08193 Bellaterra, Spain; 3Departament de Genètica i de Microbiologia, Universitat Autònoma de Barcelona, 08193 Bellaterra, Spain; 4Nanoligent S.L., Eureka Building, Av. de Can Doménech s/n, Campus de la UAB, 08193 Bellaterra, Spain; olivia.cano.garrido@gmail.com (O.C.-G.); naroas@nanoligent.com (N.S.); mcano@nanoligent.com (M.C.); mrm@nanoligent.com (M.R.-M.); 5Departamento de Química, Cátedra de Química Biológica, Facultad de Ciencias Exactas, Físicas y Naturales, ICTA, Universidad Nacional de Córdoba, Av. Vélez Sársfield 1611, Córdoba 5016, Argentina; 6Instituto de Investigaciones Biológicas y Tecnológicas (IIByT), CONICET-Universidad Nacional de Córdoba, Córdoba 5016, Argentina; 7Biomedical Research Institute Sant Pau (IIB Sant Pau), Sant Antoni Ma Claret 167, 08025 Barcelona, Spain

**Keywords:** nanomaterials, protein stability, nanomedicine, mutagenesis

## Abstract

Under the need for new functional and biocompatible materials for biomedical applications, protein engineering allows the design of assemblable polypeptides, which, as convenient building blocks of supramolecular complexes, can be produced in recombinant cells by simple and scalable methodologies. However, the stability of such materials is often overlooked or disregarded, becoming a potential bottleneck in the development and viability of novel products. In this context, we propose a design strategy based on in silico tools to detect instability areas in protein materials and to facilitate the decision making in the rational mutagenesis aimed to increase their stability and solubility. As a case study, we demonstrate the potential of this methodology to improve the stability of a humanized scaffold protein (a domain of the human nidogen), with the ability to oligomerize into regular nanoparticles usable to deliver payload drugs to tumor cells. Several nidogen mutants suggested by the method showed important and measurable improvements in their structural stability while retaining the functionalities and production yields of the original protein. Then, we propose the procedure developed here as a cost-effective routine tool in the design and optimization of multimeric protein materials prior to any experimental testing.

## 1. Introduction

The acquisition and grouping of desired biological activities, which can be combined in a single polypeptide chain through multidomain recruitment, allows producing modular proteins with specific functionalities that are not occurring in nature [1,2,3,4,5,6]. Designing multidomain proteins is a strategy especially appealing in the generation of protein materials at micro- and nanoscales, in which the building blocks must gain assembling abilities for regulatable oligomerization [7,8]. Under urgent demands from biomedical fields such as drug delivery [9,10,11] and regenerative medicine [12,13], emerging protein materials offer the possibility to assemble as supramolecular structures of defined geometries and mechanical properties, combined with desired biological functionalities [14]. For instance, protein scaffolds in tissue engineering provide mechanically stable substrates for cell growth but also display integrin-binding peptides [15] or release growth factors [16] to enhance either substrate colonization or/and proliferation. In drug delivery, protein nanoparticles offer a nanoscale size that is convenient for the prolonged circulation of payload drugs in blood and for optimal tissue and cell penetrability [17], plus specific cell targeting through solvent-display of homing peptides (i.e., specific binders of cell surface tumoral markers [18]). However, while functional recruitment through domain fusion can be organized on the simple basis of requested bio-physical properties [4,19], the whole functionality, stability, and productivity of the target proteins in cell factories are often neglected in the first design schemes. A poor or limited yield or conformational quality of recombinant proteins (being both common events [20,21]) might result in failing materials, which despite showing potential in the clinical field, are not robust enough to be brought to the industry.

In this context, rational site-directed protein engineering allows functional and structural refinement of protein functionalities that had already been achieved by domain recruitment through gene fusion. Such fine-tuning process should consider all functional and structural aspects that could be associated with the final product, with particular emphasis on the biofabrication step. The generation of structured, hierarchical tuning processes beyond the initial modular design would ensure the robust applicability of clinically relevant products, which upon a first straightforward design, could not reach optimal performance. Driven by the need to improve current protein scaffolds available for targeted drug delivery, we set out to develop a rational refining procedure described herein with transversal applicability for any kind of protein material. Specifically, we illustrate this process with the generation of a series of humanized protein nanoparticles for their application in selective drug delivery for oncology. Protein assembling, stability, and the precise cell targeting of the whole functional nanoparticles have been polished through site-directed mutagenesis of a modular polypeptide based on the human nidogen [22] empowered with T22, a peptidic ligand of the tumoral marker CXCR4 [23,24]. The architecture of the nanoparticles formed by this protein [25] and related fusion proteins [26,27,28,29,30,31,32,33] is supported by the coordination of divalent cations such as Zn^2+^ with histidine-rich domains [34], which overhang from the building blocks [35]. The rational selection of mutational targets and the sequential assay-and-error application of semi-rational protein design has rendered a fully tailored polypeptide that, in contrast to the parental protein, fulfills all the requirements for clinical development, including a smooth biofabrication process.

## 2. Materials and Methods

### 2.1. Rational Refining Procedure

The refining methodology consisted of the application of successive prediction algorithms to the structural data of the desired protein material. This was implemented in two stages. In the first one, the protein structure was subjected to a stability analysis to pinpoint areas contributing negatively to the overall robustness of the product, either by promoting precipitation or displaying excessive thermal mobility. Both issues were tackled in parallel by comparing precipitation and B-factor score predicted by Aggrescan3D 2.0 [36] and ResQ (via I-TASSER) [37,38] algorithms, respectively. High positive values in Aggrescan3D were the main indicator of instability, and residues harboring such scores were preselected for refinement, especially if they were consistent with a peak in B-factor.

In the second stage, the impact of each preselected mutation was estimated while taking into account all possible residue substitutions. The implementation of this step was performed by combining the output of DeepDDG [39], a neural network-based algorithm used to predict stability changes of protein point mutations, and STRUM [40], a machine learning algorithm trained on evolutionary information. In both stages, comparison of the scores between any of the used algorithms required a Z-score normalization of the data to cope with differences in scale of the outputs. High positive values of each index were favorable for choosing a given mutation. In the final decision regarding which residues would serve as a replacement, hydrophobic amino acids were not considered as suitable for surface-exposed sites to avoid protein insolubility and aggregation. Reciprocally, the introduction of polar amino acids should be avoided in buried residue positions.

### 2.2. Protein Production and Characterization

Proteins were designed in-house, and the encoding genes were provided by GeneArt (Thermo Fisher, Waltham, MA, USA) subcloned into pET26b plasmids (Novagen-Merck, Darmstadt, Germany). The parental protein was T22-HSNBT1-H6, and all its derivatives were named T22-HSNBT2-H6 through T22-HSNBT8-H6. Protein-encoding plasmids were transformed into *Escherichia coli* BL21 DE3 (Novagen-Merck, Darmstadt, Germany), and the encoded protein was produced overnight (O/N) at 20 °C in Lysogeny Broth (LB) medium upon induction with 0.1 mM isopropyl-β-D-1-tiogalactopyranoside (IPTG). Cells were then harvested by centrifugation (15 min at 5000× *g*) and resuspended in wash buffer (20 mM Tris, 500 mM NaCl, 10 mM Imidazole, pH 8) in the presence of protease inhibitors (cOmplete™ EDTA-Free, Roche, Basel, Switzerland). Cells were then disrupted in an EmulsiFlex-C5 system (Avestin, Ottawa, ON, Canada) by 3 rounds at 8000 psi. The soluble fraction of the cell lysate containing the proteins was separated by centrifugation (45 min at 15,000× *g*) and then charged into a HisTrap HP column (GE Healthcare, Chicago, IL, USA) for purification by immobilized metal affinity chromatography (IMAC) in an ÄKTA pure system (GE Healthcare). Protein elution was achieved by applying a linear gradient of elution buffer (20 mM Tris, 500 mM NaCl, 500 mM Imidazole, pH 8). Purified protein fractions were then dialyzed against sodium carbonate (166 mM NaCO_3_H, pH 8). Protein purity was determined by SDS-PAGE gel electrophoresis and subsequent Western blot immunodetection using an anti-His monoclonal antibody (Santa Cruz Biotechnology, Dallas, TX, USA). Protein integrity was also determined by MALDI-TOF mass spectrometry. The final protein concentration was determined by Bradford assay and Nanodrop.

### 2.3. Morphometric Characterization and Zeta Potential

Protein nanoparticle sizes were measured by Dynamic Light Scattering (DLS) at 633 nm prior to and after Zn-mediated oligomerization. Volume size distribution (expressed in nm) and zeta potential (expressed in mV) of protein materials were determined in a Zetasizer Pro (Malvern Instruments, Malvern, United Kingdom). All samples were measured in triplicates, and protein concentration was set at 1.5 mg/mL in carbonate buffer (166 mM NaCO_3_H, pH 8, refractive index 1.33, viscosity 0.887 mPa·s).

### 2.4. Determination of Intrinsic Fluorescence and Unfolding Temperature

Fluorescence spectra were recorded in a Cary Eclipse spectrofluorometer (Agilent Technologies, Santa Clara, CA, USA). A quartz cell with a 10 mm path length and a thermostatted holder was used. The excitation and emission slits were set at 5 nm. The excitation wavelength was set at 295 nm, and the emission spectra were acquired within the 310–450 nm range. The protein concentration was 0.2 mg/mL in carbonate buffer (166 mM NaHCO_3_, pH 8). In order to evaluate the conformation stability against heating, we obtained the fluorescence spectra at 5 °C temperature steps, and we calculated the center of spectral mass (CSM) for each spectrum. CSM is a weighted average of the fluorescence spectrum peak. It is also related to the relative exposure of the tryptophan (Trp) in the protein to the environment [41]. The maximum red-shift in the CSM of the Trp is compatible with large solvent accessibility and the protein unfolding. The CSM was calculated for each of the fluorescence emissions according to the following equation, where Ii is the fluorescence intensity measure at the wavelength λ*i*.
$$\lambda =\frac{\sum \lambda_{i}\cdot I_{i}}{\sum I_{i}}$$

Midpoint unfolding temperature (Tm) was determined in GraphPad Prism 8.0.2 as the temperature value that corresponds to the inflexion point of the CSM vs. temperature curve.

### 2.5. Monomethyl Auristatin E Conjugation

T22-HSNBT1-H6, T22-HSNBT2-H6, and T22-HSNBT5-H6 through T22-HSNBT8-H6 were covalently linked to maleimide-functionalized Monomethyl auristatin E (MC-MMAE, 911 g/mol, Levena Biopharma, San Diego, CA, USA) through solvent-exposed cysteine-tiol and lysine-amine groups at a 1:10 protein:MMAE molar ratio upon incubation for 4 h at RT in a one-pot reaction [29]. The concentration of each protein was set to 1.5–2 mg. Afterward, all protein conjugates were dialyzed (4 rounds of 90 min) against sodium carbonate buffer in 12–14 kDa MWCO membranes (Spectra/Por™ 2) to remove unlinked MMAE and centrifuged at 15,000× *g* for 15 min to remove precipitated protein. Reaction efficacy was finally checked by MALDI-TOF mass spectrometry.

### 2.6. Protein Modeling and Statistical Analyses

Visualization and modeling of the three-dimensional structure of T22-HSNBT-H6 were conducted in ChimeraX software (version 1.2), and molecular lipophilicity potential was calculated using pyMLP [42,43]. Gaussian distribution of values was not assumed for any of the compared datasets. Hence, the nonparametric Mann–Whitney U test was used to compare means and extract significance values. All statistical analyses were performed using the built-in analysis tool in GraphPad Prism version 8.0.2.

## 3. Results and Discussion

### 3.1. In Silico Procedure

The presented rational refining procedure is herein demonstrated in oligomeric protein nanoparticles envisaged for nanomedical applications, namely the cell-targeted delivery of antitumor payload drugs. These nanoparticles result from the organization of the modular protein T22-HSNBT1-H6 with the assistance of Zn cations present in the media (Figure 1A, Patent WO2021130390). This protein is derived from the humanized version of the CXCR4-directed T22-GFP-H6 [25,44] that was re-designed, substituting GFP by the G2 domain of nidogen (Uniprot identifier: P14543) [25].

Implementation of the first step of the refining process yielded an aggregation/flexibility profile (Figure 1B) that was used to pre-select nine sites of interest to be mutated (V45, V121, F157, V176, I200, Y201, V215, V236, and L237). The first 25 residues of the protein were not eligible for mutation as they contain the ligand peptide T22 and a flexible spacer required for proper CXCR4 targeting. Four of the preselected residues were finally selected after comparison of the predicted aggregation propensity, flexibility (ResQ, B-factor), and ΔΔG (STRUM) scores, all stacked together in Figure 1C. Then, in the second step of the process, all possible amino acid substitutions with the potential to improve stability and solubility of the protein material were evaluated by combining ΔΔG scores of both STRUM and DeepDDG methods in a normalized data plot (Figure 1D). All four chosen mutations (V45T, V121Q, F157E, and V215T) involved residues exposed on the surface of the assembled protein material. Due to the hydrophilic nature of the new residues, the predicted lipophilicity was expected to decrease in comparison to the original material (Figure 1E,F). Single mutants were designed to encompass each of the best-scoring substitutions individually, and two additional mutants were designed to harbor three and all four of the selected mutations. The rationale behind the mutant with three out of the four possible mutations (T22-HSNBT6-H6) was to include as many mutations as possible without compromising the targeting properties given by T22. Then, the mutation Val215, which is the closest to T22, was not included in this construct. A modular view of the proteins and their incorporated mutations is presented in Figure 2.

An additional clone (T22-HSNBT8-H6) was also designed to incorporate all the selected mutations plus an additional substitution at C214 to remove the capacity to bind MMAE at that particular site, which was suspected of inducing protein precipitation. As this residue was already identified, only the second step of the procedure was applied to choose a substitute residue (serine). Obtaining this additional mutant was highly interesting to reduce the potential oligomeric states of the protein, envisaging highly soluble monomers still capable of forming nanoparticles upon coordination with cationic Zn [45].

In this context, we selected the I-TASSER server as the starting point of the process because, aside from integrating the ResQ algorithm, it outputs a PDB file with the predicted protein structure of the query sequence, which can be used directly as input for Aggrescan3D. Interestingly, the feasibility of using low-resolution structure models for high-accuracy stability change prediction upon point mutations has already been demonstrated [46], and a reliable tridimensional model is not required. In fact, some sequence-based methods exploiting evolutionary information show performances comparable to structure-based tools [46]. Nevertheless, new algorithms for the determination of protein stability after point mutations are constantly under development and could theoretically be used indistinctively to reproduce the current strategy. Several methods propose diverse approaches to the same end, allowing sequence or structure-based inputs [47], the incorporation of biochemical properties data [48], or the stability prediction of multiple point mutations [49]. Needless to say, protein materials with expected enzymatic or binding capacities are susceptible to compromises in activity to favor solubility whenever a mutation tackles a catalytic residue or binding site. In such cases, users should consider the implications of proceeding with the given mutation.

### 3.2. Protein Expression and Purification

As stated above, the modular design of T22-HSNBT1-H6 [25] is similar to T22-GFP-H6. The GFP-based version forms multimeric nanoparticles of 12 nm efficiently targeted to CXCR4+ cancer cells [44]. Several mutated versions of the parental protein (from T22-HSNBT1-H6 to T22-HSNBT8-H6) were designed and produced in E. coli according to the defined purposes described above. For simplification purposes, the terms T22-HSNBT-H6 and HSNBT will be used indistinctively from now on (as synonyms) in text and figures. An initial expression test was carried out as a first approach to evaluate the feasibility of recombinant production and the quality of each new mutant protein. All mutant proteins were successfully detected by Western blot in the cell lysates, after cell disruption (Appendix AA), with molecular weights within the expected range (~30.3 kDa). Proteins containing the mutation F157E (namely HSNBT4, HSNBT6, and HSNBT7) exhibited a slightly higher migrating band. The analysis of HSNBT3 and HSNBT4 was halted at this point because an important fraction of these proteins occurred in the insoluble cell fraction (data not shown). HSNBT8, harboring the most mutations, was designed individually after verifying that its closest construct, HSNBT7, was properly produced. At this point, protein production was scaled up to 500 mL cultures maintaining the same conditions. All the proteins were successfully purified by metal affinity chromatography with suitable yields (Appendix AB), and their integrity was assessed by both SDS-PAGE/Western blot (Appendix AC) and MALDI-TOF (Appendix AA), being in 166 mM NaCO_3_H buffer at pH 8. Mass spectrometry profiles confirmed that the mobility shift in SDS-PAGE of proteins harboring the mutation F157E (HSNBT6, HSNBT7, and HSNBT8) was not caused by any unexpected insertion (molar mass did not vary) and was rather due to conformational changes and/or changes in SDS binding induced by the newly accommodated glutamic acid.

### 3.3. Stability Study, Size Distribution, and Nanoparticle Formation

Several indicators of stability were relied upon to evaluate the properties of the new protein materials [50,51]. First, zeta potential (Figure 3A) was measured to evaluate the degree of repulsion between proteins in the dispersion and have a grasp of their stability. Generally, it is considered that high (either positive or negative) values of zeta potential are indicative of suitable stability of the system [52]. In this case, negative zeta potentials were obtained for all proteins, and while no significant differences were seen, a trend was evident in which all the mutated candidates exhibited more negative averages compared to the original protein HSNBT1. Regardless, a significant change in zeta potential was not expected as surface charges were mostly unaltered, and only one negative charge was introduced (F157E).

Second, the intrinsic fluorescence of the proteins was used to assess unfolding parameters linked to temperature [53,54]. We found that the Tm of the mutant proteins (Figure 3B) was in the same range as the original material yet slightly higher, indicative of increased tolerance and stability against increasing temperature. Since, among other uses, protein materials are particularly useful as drug carriers, this ability was explored by chemical conjugation with MMAE, a potent antitumor drug. Due to the hydrophobic nature of MMAE, it is essential to ensure the solubility of the conjugate, which was evaluated by assessing protein precipitation in the post-conjugation dialysis with the storage buffer (166 mM NaCO_3_H, pH 8), presenting the data as a percentage soluble protein remaining relative to the original concentration (Figure 3C). Significant improvements were reported for all mutants (above 85% solubility), except for HSNBT6, which showed high variability in replicate measurements, still remaining far more soluble on average than the original material. Additionally, the cysteine removal in HSNBT8 greatly contributed to the stabilization of the protein during conjugation, as no precipitation was detected in the assays involving this candidate. The maleimide functional group in MMAE preferably reacts with thiol groups in cysteines. Then, the only accessible cysteine not forming disulfide bonds was Cys214 (HSNBT1-HSNBT7), so conjugation at this position was deemed destabilizing. Instead, HSNBT8 had Cys214 replaced with Ser214, meaning that MMAE was distributed throughout the many amine groups from exposed lysines, keeping the protein material soluble. Supporting this claim, up to one linked MMAE was detected via MALDI-TOF in the soluble fractions of HSNBT1-HSNBT7, while up to three linked MMAE could be identified in HSNBT8, which did not precipitate (Appendix AD).

At this point, HSNBT6 and HSNBT8 were selected to proceed further with the characterization because of their high production yields (namely 26 and up to 28 mg/L, respectively). Volume size distribution was studied via DLS using HSNBT1 as a reference. As depicted in Figure 3D, the same size was maintained in all protein materials, confirming that the chosen mutations did not affect the quaternary structure of the protein in its native state.

Finally, the capacity of the new protein materials to assemble into nanoparticles was explored by adding ZnCl_2_ at a final concentration of 0.04–0.08 mM. In all cases, an increase in size was achieved (Figure 3D), reflecting nanoscale oligomerization. Specifically, HSNBT6 and HSNBT8 formed nanoparticles of around 17.5 and 11.5 nm, respectively, while the reference HSNBT1 yielded 31 nm-nanoparticles. Smaller nanoparticles in HSNBT8 were expected, as the lone cysteine at position 214 that was removed could promote the dimerization of the building blocks before assembly with Zn^2+^, leading to bigger nanoparticles. Nevertheless, these size ranges were within those considered optimal in medical applications, namely between 10 and 100 nm, that favor recirculation in the bloodstream and prevent renal clearance [55]. Most importantly, the results also confirm that mutations that originated from the refining protocol did not interfere with the Zn-driven assembling capacity of the building block. Hence, both candidates HSNBT6 and HSNBT8 represent successful improvements over the starting material, suitable for clinically oriented applications.

## 4. Conclusions

In this study, we have presented a sequential in silico screening procedure to be routinely applied to improve the stability of nanoscale protein materials envisaged for biomedical applications. This platform provides an array of favorable mutations prone to stabilize the final product, either in the monomeric form or as nano-sized oligomers. While of a non-automated nature, the method offers flexibility and possibilities for further modulation since the users may follow the basic steps using their preferred algorithms, fitting specific needs or improving the whole process with updated algorithms. Thus, the inclusion of brief in vitro optimization methods, such as the one described herein, into protein design processes should improve the success chance of the resulting materials at virtually no cost. Moreover, adapting such methodologies will enhance project viability prior to reaching the experimental phase, where having unsuccessful candidates translates into large expenditures of time, personnel, and laboratory consumables.

## 5. Patents

Protein scaffolds described herein are protected under patent WO2021130390A1, licensed to Nanoligent S.L.

## Figures and Tables

**Figure 1 ijms-23-04958-f001:**
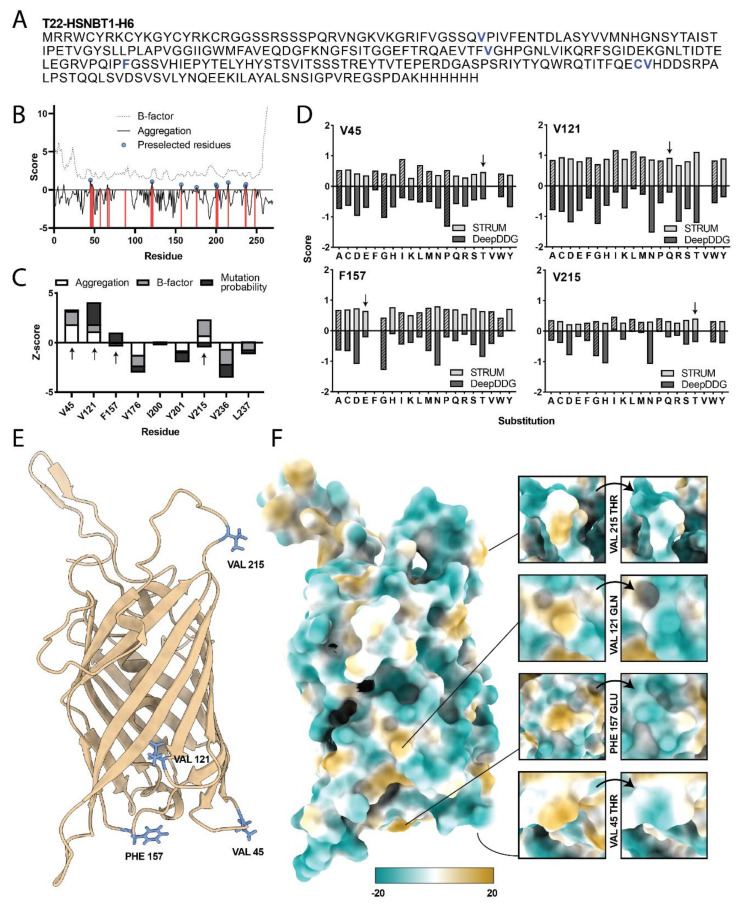
(**A**) Amino acid sequence of the construct T22−HSNBT1−H6. Residues mutated at some point are highlighted in bold and blue; numeration starts at Met0. (**B**) Visualization of B-factor (dotted line) and aggregation (continuous line) predictors for each residue in T22-HSNBT1-H6. Positive aggregation values indicate tendency to aggregate and are highlighted in red. Preselected residues are marked with blue dots. (**C**) Stacked aggregation (Aggrescan3D), B-factor (ResQ), and mutation probability (STRUM) prediction scores of the preselected residues. (**D**) Scores of STRUM (light gray) and DeepDDG (dark gray) predictors for each possible substitution in the four selected residues. Hydrophobic residues are filled in a diagonal hatch pattern, and arrows indicate the chosen substitution for each amino acid. (**E**) Three-dimensional representation of T22-HSNBT1-H6, with the atoms of the selected residues for mutation displayed in blue. (**F**) Surface lipophilicity potential map. Insets on the right side highlight the predicted lipophilicity of the selected residues before and after substitution. Coloring ranges from dark cyan (most hydrophilic) to white to dark gold (most lipophilic).

**Figure 2 ijms-23-04958-f002:**
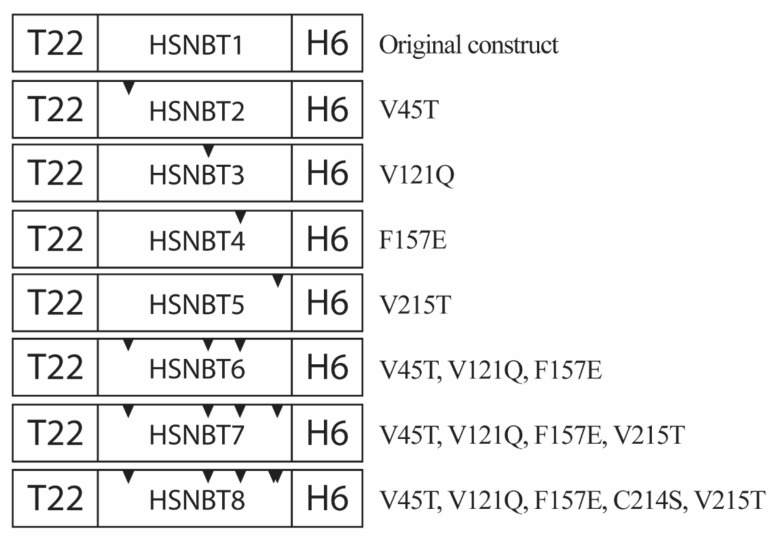
Modular design of the proteins derived from T22-HSNBT1-H6. Inverted black triangles indicate the approximate location of each chosen mutation in the core module of the scaffold protein. The precise position and nature of each mutation are indicated next to the modular construct, providing the original residue, its position in the sequence, and the new residue.

**Figure 3 ijms-23-04958-f003:**
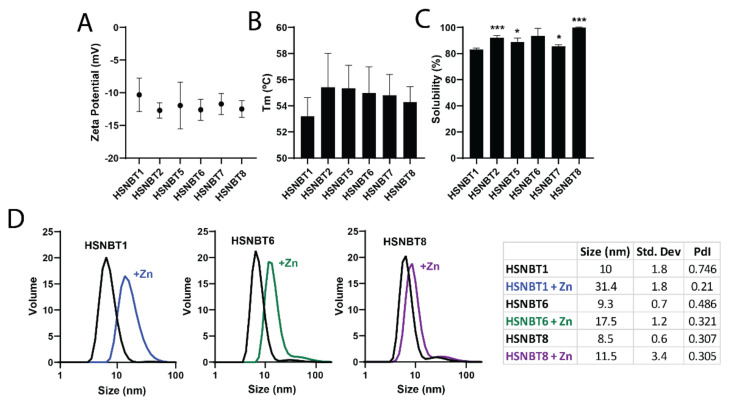
(**A**). Distribution of the zeta potential values measured for the studied proteins. (**B**) Tm of the studied proteins obtained from CSM curves. (**C**) Precipitated portion of each of the protein mutants after the conjugation with the drug MMAE. (**D**) Size-volume profile (**left**) and size data (**right**) of selected candidates measured by DLS before and after nanoparticle formation induced with ZnCl_2_. Significance indicated as: * *p* ≤ 0.05, *** *p* ≤ 0.001.

## Data Availability

The data presented in this study are openly available at: https://www.dropbox.com/s/p5h0ogt5cf5t0rv/DATA.xlsx?dl=0 (accessed on 1 April 2022).

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
