# Peer review of "An In Silico Methodology That Facilitates Decision Making in the Engineering of Nanoscale Protein Materials"

_ijms, 2022, doi:10.3390/ijms23094958_

Round 1

Reviewer 1 Report

Dear Authors;

Re: Manuscript ID: ijms-1686926
Type: Communication
Title: An in silico methodology that facilitates decision-making in the engineering of nanoscale protein materials.

The paper describes a sequential in silico screening procedure to be routinely applied to improve the stability of nanoscale protein materials envisaged for biomedical applications. 

Please consider the following comments / suggestions:

1. With respect of thermodynamic principles and biophysics the expression "self- assembling" / "self-assembly" should either be:

justified adequately

or

removed.

2. You mentioned: "... procedure developed here as a cost-effective routine tool in the design and ...". How did you come to this point? Please describe.

3. The sentence in Lines 54-56 " ... that even clinically promising and with potential usability, ..." needs to be re-written. 

4. Medium viscosity and refractive index need to be specified for Particle Size analysis.

5. In Line 70 you mentioned: "...nanoparticles formed by this ...". Please specify "this".

6. Size changes must be presented clearly, e.g. by using Tabulation. 

7. Statistical analysis need to be described in more details.  

Author Response

Please consider the following comments / suggestions:

  1. With respect of thermodynamic principles and biophysics the expression "self- assembling" / "self-assembly" should either be:

justified adequately or removed.

Thank you for pointing out this issue. Certainly, as you indicate, the type of assembly that we describe in this work is derived from the coordination of zinc with histidine residues in the protein, rather than proper self-assembly driven, for example, by interacting surface motifs. Accordingly, we have removed the term “self-assembly/ing” from the manuscript as it was not properly used.

  1. You mentioned: "... procedure developed here as a cost-effective routine tool in the design and ...". How did you come to this point? Please describe.

Thank you for your comment. In protein engineering, researchers will usually choose diverse domains or ligands of interest to design a fusion protein that accommodates traits of interest, inherited from different original proteins. However, when parts of different proteins are fused, one cannot be sure if the final folded state of the new protein will expose residues that were originally buried in the source proteins and would negatively affect the overall solubility and stability. Instead of finding out that our protein is non-viable in the wet lab, having invested time, staff and consumables to produce/purify it, we propose the implementation of our in silico methodology because it comes virtually at near-zero associated cost, especially compared to the burden of having to re-design, synthesize, produce and purify new candidates.

As you pointed out, this reasoning was poorly reflected in the manuscript, so we have rewritten the conclusions to address it.

  1. The sentence in Lines 54-56 " ... that even clinically promising and with potential usability, ..." needs to be re-written. 

Yes, thank you. We have rewritten the sentence to make it clearer.

  1. Medium viscosity and refractive index need to be specified for Particle Size analysis.

Certainly, we have added this missing data in the “morphometric characterization and zeta potential” section of the materials and methods. Thank you.

  1. In Line 70 you mentioned: "...nanoparticles formed by this ...". Please specify "this".

Yes, apologies, the word “protein” was missing in the text and has now been added. Thank you for pointing it out.

  1. Size changes must be presented clearly, e.g. by using Tabulation. 

Thank you for the observation, we have now provided a new version of figure 3,  which contains a clear table to adequately display size data.

  1. Statistical analysis need to be described in more details.  

Thank you, yes, we have updated the materials and methods section 2.6 to provide a more detailed description of the criteria and statistical tests used to compare means.

Reviewer 2 Report

Manuscript submitted by Parladé et. al "An in silico methodology that facilitates decision-making in the engineering of nanoscale protein materials" in International Journal of Molecular Sciences (ijms-1686926). I have been carefully read the manuscript and these points should be addressed in the manuscript, as explained below:

  1. What is the motive behind the performed study? It should be included in the introduction section.
  2. Figure 3: visibility is not good. Author can make the more visible image and supply in the manuscript.
  3. Page3, 134-135: It is also relatedto the relative exposure of the tryptophan (Trp) in the protein to the environment. To support this sentence with literature the author should cite the article: Int J Biol Macromol. 2015, 72, 875-882.
  4. Page 5, line 240-241: “Several indicators of stability were relied upon to evaluate the properties of the new protein materials”. To support this sentence with literature the author should cite the article: “Cell Biochem Biophys. 2012, 62 (3), 487-499”. “Appl Microbiol Biotechnol. 2014, 98 (6), 2533-2543”.
  5. The last sentence of "conclusion" needs revision.
  6. Page 6, line 251-252: “Second, the intrinsic fluorescence of the proteins was used to assess unfolding parameters linked to temperature”. To support this sentence with literature the author should cite the article: “Cell Biochem Biophys. 2011; 61(3):551-560”. RSC Advances 2015, 5 (26), 20115-20131.
  7. What were the rational in DLS experiments?

Author Response

1. What is the motive behind the performed study? It should be included in the introduction section.

Yes, thank you for the question. This study originated from the need to develop a new version of our humanized scaffold (HSNBT). We had concerts regarding thermal stability and compatibility with chemical conjugation, so we set out to define a method by which any protein could be improved in terms of stability, regardless of its intended use.

As suggested, we have included our motivation in the introduction section.

2. Figure 3: visibility is not good. Author can make the more visible image and supply in the manuscript.

Thank you for pointing out this issue, we have prepared a new figure to make data more visible for readers.

3. Page3, 134-135: It is also relatedto the relative exposure of the tryptophan (Trp) in the protein to the environment. To support this sentence with literature the author should cite the article: Int J Biol Macromol. 2015, 72, 875-882.

Thanks, the reference has been added.

4. Page 5, line 240-241: “Several indicators of stability were relied upon to evaluate the properties of the new protein materials”. To support this sentence with literature the author should cite the article: “Cell Biochem Biophys. 2012, 62 (3), 487-499”. “Appl Microbiol Biotechnol. 2014, 98 (6), 2533-2543”.

Both references have been included, thank you for the recommendation.

5. The last sentence of "conclusion" needs revision.

Thank you, we have rewritten the final part of the conclusions accordingly.

6. Page 6, line 251-252: “Second, the intrinsic fluorescence of the proteins was used to assess unfolding parameters linked to temperature”. To support this sentence with literature the author should cite the article: “Cell Biochem Biophys. 2011; 61(3):551-560”. RSC Advances 2015, 5 (26), 20115-20131.

Both references have been included, thank you for the recommendation.

7. What were the rational in DLS experiments?

Thank you for the question. As we generally work with proteins for cancer treatment, we have a special interest in proteins that have the ability to assemble into higher order structures (typically of toroid shape) sized between 10-100 nm. In this size range they are able to escape renal filtration and remain in circulation more time in the bloodstream, requiring low amounts of administered drug in in vivo experiments. Dynamic light scattering (DLS) is really useful to us to study the size of our proteins before and after nanoparticle assembly. The technique is compatible with our most of the storage buffers we employ and the addition of ZnCl2 does not affect its results. For this reason, we use DLS in the characterization of the obtained proteins after the refining process described in the manuscript.

We have clarified the usage of DLS in the Materials and Methods.